# Aluminum Concentration Is Associated with Tumor Mutational Burden and the Expression of Immune Response Biomarkers in Colorectal Cancers

**DOI:** 10.3390/ijms252413388

**Published:** 2024-12-13

**Authors:** Rita Bonfiglio, Erica Giacobbi, Valeria Palumbo, Stefano Casciardi, Renata Sisto, Francesca Servadei, Maria Paola Scioli, Stefania Schiaroli, Elena Cornella, Giulio Cervelli, Giuseppe Sica, Eleonora Candi, Gerry Melino, Alessandro Mauriello, Manuel Scimeca

**Affiliations:** 1Department of Experimental Medicine, Tor Vergata Oncoscience Research (TOR), University of Rome “Tor Vergata”, 00133 Rome, Italy; rita.bonfiglio@uniroma2.it (R.B.); erica.giacobbi@gmail.com (E.G.); valeria.palumbo95@hotmail.com (V.P.); francescaservadei@gmail.com (F.S.); sciolimp@hotmail.it (M.P.S.); schiaroli@yahoo.com (S.S.); candi@uniroma2.it (E.C.);; 2Department of Occupational and Environmental Medicine, Epidemiology and Hygiene, INAIL Research, Monte Porzio Catone, 00078 Rome, Italy; s.casciardi@inail.it (S.C.); r.sisto@inail.it (R.S.); 3Department of Surgery, Tor Vergata Oncoscience Research (TOR), University of Rome “Tor Vergata”, 00133 Rome, Italy

**Keywords:** aluminum, colorectal cancer, immune escape, immune checkpoint, IFNγ, myeloid cells, environmental pollution

## Abstract

Environmental pollution poses a significant risk to public health, as demonstrated by the bioaccumulation of aluminum (Al) in colorectal cancer (CRC). This study aimed to investigate the potential mutagenic effect of Al bioaccumulation in CRC samples, linking it to the alteration of key mediators of cancer progression, including immune response biomarkers. Aluminum levels in 20 CRC biopsy samples were analyzed using inductively coupled plasma mass spectrometry (ICP-MS). The results indicated that Al bioaccumulation occurred in 100% of the cases. A correlation between Al levels and tumor mutation burden was observed. Furthermore, RNA sequencing revealed a significant association between Al concentration and the expression of the immune checkpoint molecule CTLA-4. Although correlations with PD-1 and PD-L1 were not statistically significant, a trend was observed. Additionally, a correlation between Al levels and both the presence of myeloid cells and IFNγ expression was detected, linking Al exposure to inflammatory responses within the tumor microenvironment. These findings suggested that Al can play a role in CRC progression by promoting both genetic mutations and immune evasion. Given the ubiquitous presence of Al in industrial and consumer products, dietary sources, and environmental pollutants, these results underscored the need for stricter regulatory measures to control Al exposure.

## 1. Introduction

Environmental pollution is a major concern for human health, with toxic metals posing a significant public health risk worldwide [1]. These metals contribute to water, soil, and air pollution, originating from a variety of industrial, household, and agricultural activities [2]. Acute and chronic exposure to toxic metals has been associated with the development of pathological conditions [3,4]. These metals frequently have mutagenic capabilities [5] and are also involved in the dysregulation of oxidative stress [6], inflammation [7], and protein folding [8].

A common metal to which people are chronically exposed is aluminum (Al). Beyond its natural distribution in soil, high environmental levels of Al are typically due to mining and processing activities aimed at producing Al compounds. These activities involve the extraction of bauxite ore, followed by refining processes that release significant amounts of Al into the environment [9]. Al is ubiquitous in industrial products such as beverage cans, pots and pans, airplanes, siding and roofing, foil, and cosmetics [10]. The use of Al in these products is driven by its advantageous properties, including light weight, resistance to corrosion, and high thermal conductivity, making it essential in various manufacturing sectors.

Due to its widespread use, Al exposure can occur via multiple routes including inhalation, ingestion through food and beverages, and dermal absorption. Al compounds are frequently used as food additives, such as stabilizers, colorants, and leavening agents, particularly in processed foods [11,12]. Additionally, the use of Al cookware and utensils during food preparation can increase its content in meals, particularly when cooking acidic or salty foods. Beverages such as tea [13], which naturally contain Al, and Al-containing packaging materials further contribute to dietary exposure.

The daily intake of Al from food is estimated to range from 1 to 10 mg/day in the general population [14], though higher levels are observed in individuals consuming processed or packaged foods. This extensive exposure can lead to significant accumulation within the organism, potentially affecting health.

Mechanisms underlying Al-related organ toxicity are generally associated with reactive oxygen species (ROS) production resulting from increased oxidative stress and inflammation [15]. Oxidative stress associated with long-term exposure to Al can lead to chronic inflammation [16], a known risk factor for various diseases, including cancer [17]. In fact, Al can induce an immunosuppressive microenvironment by dysregulating immune checkpoint expression, thus supporting the survival, proliferation, and invasion of cancer cells [18]. Regarding the accumulation of Al in colon tissues, an epidemiological study revealed no significant differences in its concentrations by analyzing trace elements in both healthy and CRC biopsies [19]. However, the Al levels were slightly higher in CRCs compared to normal ones. In this context, we previously detected Al accumulation in colorectal cancer (CRC) biopsies using histopathological techniques [20], establishing a link between Al bioaccumulation and key biological processes associated with CRC progression, such as epithelial-to-mesenchymal transition (EMT) [21,22] and resistance to cell death [23,24,25,26]. Additionally, a multiomics analysis conducted on two samples further suggests a correlation between Al presence and the occurrence of DNA mutation events. The presence of Al in CRC, the pathogenesis of which is rather complex [27,28,29,30], could be related to the high frequency of Al contamination of food; therefore, intestinal absorption could be considered as a primary route of Al exposure, mainly considering its bioaccumulation in the gastrointestinal tract. To further understand whether Al toxicity is dependent on its concentration within samples, in the present study, we performed ICP-MS analysis to detect Al in 20 CRC biopsies subjected to both mutational and RNAseq analyses. By analyzing the levels of Al in cancerous tissues and correlating these levels with tumor mutational burden (TMB) and immune evasion markers, the study seeks to uncover whether Al exposure can be considered a prognostic factor for patients affected by CRC. Therefore, here, we provide insights into the role of environmental pollutants, such as Al, in influencing the genetic and immunological landscape of CRC.

## 2. Results

### 2.1. Clinical CRC Cohort Investigated and Al Detection

CRC samples derived from male and female patients were retrospectively collected (Table 1). Ten samples derived from right colon dx, whereas 10 originated from cancer in the left colon. The tumor sizes ranged from 2.3 cm to 10 cm. Histological examination showed that all the analyzed lesions were adenocarcinoma (Figure 1A,B): 2/20 low-grade adenocarcinomas and 18/20 high-grade adenocarcinomas. Tissue infiltration was observed in 1 low-grade adeno-carcinoma and in 18 high grade adenocarcinomas. Seven patients had lymph node metastasis at the time of surgery (Table 1).

Quantitative ICP-MS analysis for Al bioaccumulation detected 100% positivity in the investigated cases. The CRC samples showed a mean concentration of 15.0 ± 1.52 mg/kg of dried tissue. The maximum quantification value was 29 mg/kg, while the minimum was 5 mg/kg (Figure 1C). Morin staining showed a positivity in samples with Al concentration greater than 13 mg/k; in all cases, Al deposits were observed in tumor areas, specifically in the cancer cell cytoplasm (Figure 1D).

### 2.2. Mutational State and Al Accumulation

The possible mutational effect of Al bioaccumulation has been evaluated by correlating the Al levels with the TMB of colon cancer lesions. Figure 2A showed that results from linear regression analysis indicated a significative positive correlation with TMB (*p* < 0.0001; R-squared = 0.86). The mutational analysis also showed numerous somatic mutations in hallmark genes in the patient with the higher Al concentration (Table 2). Although the Al concentration appears to be positively associated with the TMB, the mutational analysis did not reveal recurrent mutations in specific genes.

### 2.3. Al Bioaccumulation and Colon-Cancer-Related Immune Response

RNA-seq data concerning the IFNγ expression were also correlated with the Al accumulated amount. Linear regression analysis highlighted a positive correlation between IFNγ levels within colon cancer lesions and their corresponding Al concentration (*p* = 0.0002; R-squared = 0.78) (Figure 2B).

To evaluate the possible association between Al accumulation and the immune escape ability of cancer cells, we correlated data from RNA-seq concerning CTLA4, PDL1, and PD1 and the Al amount found in the corresponding samples.

As shown in Figure 3A, linear regression analyses demonstrated a positive significant correlation between Al levels and CTLA4 expression in colon cancer lesions (*p* < 0.0001; R-squared = 0.85). For PDL1 (Figure 3B) and PD1 (Figure 3C), no significant association has been observed despite a positive trend appearing evident. Immunohistochemical analysis confirmed the high expression of CTLA4, PDL1, and PD1 by CRC cells in patients with a high level of Al (Figure 3). Multivariate analysis further confirmed the association between Al concentration and high TMB. In fact, a 3D biplot graph (see Appendix A) showed a prominent role of Al in discriminating high and low TMB samples. A similar trend was observed for IFNγ and CTLA4 (Appendix A).

RNASeq data have been used to quantify the abundance of immune and stromal cell populations in a sub-cohort of 15 CRC biopsies by using an MCPCounter [31] (Table 3). Specifically, the MCPCounter allowed us to identify the following cell types: B cell, macrophages, T lymphocytes, CD8 positive lymphocytes, cancer-associated fibroblasts, myeloid cells, and neutrophils.

It is noteworthy that Person analysis showed a moderate positive association only between Al concentration and myeloid cells (Figure 4A,B). This immune cell type is generally linked to an immunosuppressive cancer environment.

No association between Al concentration and age was observed. T-student analysis displayed the difference in Al concentration between male and female (F 17.1 ± 2.0; M 12.9 ± 2.2; *p* = 0.17) and patients with or without lymph node metastasis (no metastasis 15.31 ± 2.0; yes metastasis 14.4 ± 2.4; *p* = 0.8) (Figure 5A,B).

All association between Al concentration and continuous variables is shown in the heatmap (Figure 5C).

## 3. Discussion

This study reports an association between the concentration of Al in CRC biopsies and some hallmarks of cancer, such as high TMB and the expression of immune escape biomarkers. These findings suggest a potential role for Al in influencing tumor biology and warrant further investigation into its impact on CRC development.

The growing evidence that environmental pollution represents a critical issue for human health has led to increased efforts to investigate how pollutants impair molecular pathways involved in cancer progression. In a recent investigation, the qualitative assessment of Al bioaccumulation in CRC by morin staining has been associated to the occurrence of EMT and cell death resistance [32]. To further understand whether Al toxicity could also be dependent on its concentration within samples, in this study ICP-MS analysis was performed on 20 CRC biopsies. ICP-MS is known to be a highly performative technology, extremely useful in the detection of trace elements into histological specimens [33]. The application of ICP-MS on FFPE tissues has been recently validated in the study of Coyte et al. [34]. By comparing the ICP-MS results from FFPE and fresh tissues, the authors demonstrated that the histological preparation of tissues (formalin fixation, dehydration, and paraffin embedding) does not significantly alter the concentration of numerous bioaccumulated metals, including Al. Moreover, ICP-MS performed on fresh samples showed a significant increase in Al bioaccumulation in CRC as compared to adjacent normal mucosa [19].

It is noteworthy that ICP-MS analysis showed Al bioaccumulation on all the investigated CRC samples. The consistent presence of this toxic metal across different patients affected by CRC implies that Al exposure is a very frequent event. Possible sources of Al exposure include dietary intake, as Al is commonly found in processed foods and food additives [35]. Additionally, Al-containing antacids [36], cooking utensils [37], and packaging materials [38] could contribute to its bioaccumulation in the body. Environmental exposure is certainly another significant factor. In fact, Al has been detected in water sources, air pollution, and soil [39]. Occupational exposure in industries such as mining, manufacturing, and agriculture could also play a significant role [40,41]. Given the ubiquitous nature of Al exposure, it is crucial to characterize the molecular mechanisms by which Al exerts its toxicity primarily through slow and progressive accumulation in tissues.

In this scenario, this study reveals a positive association between Al concentration and TMB. TMB is currently considered as a prognostic/predictive factor in several cancers [42,43,44]. In fact, TMB could be a predictive indicator of response to immunotherapies, as patients with high TMB are more likely to benefit from immune checkpoint inhibition (ICI) therapies [45]. Numerous studies suggested that patients with high TMB may respond better to immunotherapy [46]. This is because the increased number of mutations can result in the formation of new antigens (neoantigens) that are more easily recognized by the immune system. Consequently, tumors with high TMB are more likely to be infiltrated by immune cells, enhancing the effectiveness of immune checkpoint blockade [47].

The clinical utility of TMB as a biomarker in CRC is still under investigation. While high TMB has shown promise in predicting responses to immunotherapy in other cancer types, its role in CRC appears complex and may depend on additional factors such as the tumor microenvironment and specific genetic alterations.

At the state of the art, there is no evidence of effects of Al on genome instability and TMB. Thus, the data reported here represent a new perspective on possible toxic effects exacerbated by Al in the human tissues. However, further investigations are required to determine if the increase in TMB could be a direct or indirect effect of Al bioaccumulation. In fact, in vitro studies have shown that Al can directly cause DNA damage [48,49]. More likely, the increase in TMB related to Al bioaccumulation could be associated with the capability of this metal to induce oxidative stress [50,51,52], interfere with DNA repair mechanisms (DRM) [53], or alter immune response [54,55].

The demonstrated ability of Al to regulate inflammatory response [56], also in the gastrointestinal tract [57], represents the common thread linking the association between Al accumulation and both high TMB and the expression of immune checkpoints, as highlighted in this study.

Aluminum may enhance monocyte recruitment and differentiation into dendritic cells (DCs) [58], inducing the production of IL-1beta and IL-18. These proinflammatory interleukins also stimulate the NLRP3 inflammasome within DCs [59,60], which in turn further increases the release of IL-1beta, enhancing both innate and adaptive responses. Moreover, Al induces the presentation of antigens to T cells, mainly CD4+ T cells, which might result in the development of a predominantly TH2 immune response [61]. Nonetheless, it has been found that long-term Al exposure significantly reduces the CD8-positive lymphocytes, thus favoring an immunosuppressive environment [62]. This suggests that while Al might induce alterations in cell immune response, the stimulating or suppressing effects could depend on the dose, route of administration, exposure duration, or cell population.

The ability of cancer to escape the immune system requires the presence of an immune-suppressive microenvironment frequently represented by the aberrant activation of immune checkpoints such as CTLA4, PD-1, and its ligand PD-1L [63,64,65,66].

RNA-seq data showed a positive correlation between Al and CTLA4 expression. Moreover, a positive trend has also been observed for PD1 and PD-L1, although no significant associations were observed. CTLA-4 is primarily expressed on the surface of T cells and acts as a negative regulator, preventing overactivation of the immune system [67]. In the context of cancer, the upregulation of CTLA-4 can contribute to immune escape mechanisms, allowing tumor cells to evade immune surveillance and promoting tumor growth and progression [68]. It is noteworthy that the data reported here also demonstrated a moderate positive association between Al and the presence of myeloid cells within the tumor microenvironment. Myeloid cells are widely recognized as key players in shaping the tumor microenvironment, significantly influencing the balance between tumor suppression and progression [69,70]. A myeloid-cell-rich tumor microenvironment is generally regarded as immunologically permissive, fostering conditions that support tumor survival and growth. These cells contribute to immune evasion by inducing the expression of immune checkpoint molecules, thereby suppressing anti-tumor immune responses [71]. Moreover, they actively promote tumor invasion and metastasis through a range of non-immunological mechanisms, further underscoring their critical role in cancer progression [71].

Although not mechanistic in nature, our findings suggest a consistent association between Al accumulation in CRC tissues and the development of an immunologically permissive tumor microenvironment. The observed correlation between IFNγ and Al, apparently paradoxical, may also align with this proposed scenario.

In fact, it is known that in the tumor microenvironment, IFNγ plays a dual role, regulating both antitumor and pro-tumorigenic immune responses. It functions as a cytotoxic cytokine, working alongside granzyme B and perforin to induce apoptosis in tumor cells [72]. At the same time, IFNγ promotes the expression of immune checkpoint and indoleamine-2,3-dioxygenase (IDO), thereby activating immune-suppressive pathways [73,74,75,76,77,78].

## 4. Materials and Methods

### 4.1. Samples Collection

A total number of 20 biopsies from FFPE samples subjected to both mutational and RNAseq analyses were retrospectively collected from patients who underwent colonic resection for sporadic CRCs. To ensure the quality of the samples, all the tissues were Hematoxylin- and Eosin (H&E)–stained and subjected to a pathological QC. The study was approved by the Institutional Ethical Committee of the “Policlinico Tor Vergata” (reference number #96-19). All the experimental procedures were conducted in accordance with the Code of Ethics of the World Medical Association, specifically the Declaration of Helsinki.

### 4.2. ICP-MS

The ICP-MS analysis was performed by Agri-Bio-Eco Laboratori Riuniti S.R.L. Briefly, four sections of 20 µm thickness were obtained from each FFPE sample. The sections were stored in 1.5 mL Eppendorf, and xylene was added and left overnight to allow paraffine melting. The xylene was then changed twice, followed by three changes of ultra-pure ethanol to completely remove paraffine residues. Thereafter, complete evaporation of ethanol was performed to fully dry the samples. The appropriately weighed samples were subjected to a digestion process using a 1:10 solution composed of hydrogen peroxide and nitric acid. The digested sample was made up to the mark of 10 mL and subjected to analysis by the ICP-MS (Agilent ICP-MS 7700, Santa Clara, CA, USA) technique. The mean quantity of the dried samples was 0.0014 mg. The concentration of Al was determined by the standard curve method. Validation parameters such as linearity, precision, and limit of detection (LOQ) were evaluated. The linearity of the calibration curves, calculated as the linear correlation coefficient R^2^, was greater than 0.998. The LOQ for Al was 3.71 µg/g. Precision values were calculated as the coefficient of variation (CV) (%) and ranged from 0.5 to 50 µg/L. The instrument employs argon gas for plasma generation and helium for operation within the collision cell; the isotope measured under these conditions was ^27^Al.

### 4.3. Morin Staining Protocol

Morin staining was performed on 4-µm FFPE sections according to Bonfiglio et al. [19]. Stained sections were observed under fluorescence microscopy (Leica DM4 M, Berlin, Germany)—to identify the presence of Al (emission peak about 510 nm, excitation 440 nm).

### 4.4. Molecular Investigation

Fresh frozen tissues were used for whole genome sequencing and RNAseq according to Yang et al. [68]. TMB was calculated as the number of non-synonymous mutations of protein coding genes divided by exome size in megabases. The abundance of immune and stromal cell populations in the cancer tissues was evaluated by using an MCPCounter on a sub-cohort of 15 CRC biopsies [31].

### 4.5. Immunohistochemistry

Immunohistochemistry was performed to investigate in situ the expression of the immune checkpoints CTLA-4, PD-1 and PD-L1 [69]. Four-µm serial sections from FFPE blocks were used for immunohistochemical studies. The sections were subjected to antigen retrieval by treating them with EDTA pH 8 for CTLA-4, PD-1, and PD-L1 using a pressure cooker (121 °C) for 15 min.

Afterwards, the sections were incubated with the following antibodies: a rabbit recombinant monoclonal anti-CTLA-4 antibody (1:500 dilution, clone CAL49, AbCam, Cambridge, UK), a rabbit monoclonal anti-PD-1 antibody (pre-diluted, CAL20, Leica), and a rabbit recombinant monoclonal anti-PD-L1 antibody (1:100 dilution, clone SP142, AbCam, Cambridge, UK) for 1 h at room temperature. Washing was performed using PBS/Tween20 pH 7.6.

The reactions were visualized using an HRP-DAB Detection Kit (UCS Diagnostic, Rome, Italy). The immuno-stained sections were observed under a light microscope (Axioscope-5, Zeiss, Oberkochen, Germany).

### 4.6. Statistical Analysis

Pearson correlation analysis was performed to evaluate the linear relationship between Al and other continuous variables. The correlation coefficient (r) and corresponding *p*-value were calculated to determine the strength and statistical significance of the associations. The analysis was conducted using Statistical software R (R version 4.2.1 (23 June 2022)).

The differences in Al concentration between categorical variables were analyzed using a *t*-test. The RNASeq data were reported as transcript per million (TPM). The Al concentration was reported as mean ± SEM.

Multivariate analyses were carried out using the statistical software Python (Python version 3.11) with relevant libraries including scikit-learn for data preprocessing and dimensionality reduction, and matplotlib for visualization. Principal component analysis (PCA) was used to explore the relationship between variables and discriminate between tissues with high or low tumor mutational burden (TMB). The PCA technique projects the data onto a subspace that maximizes variance while reducing dimensionality. The principal components are linear combinations of the original variables, capturing the largest fraction of variability. A binary variable, “TMB_index”, was introduced to classify the samples into two groups: 1 for TMB values greater than or equal to 10 and 0 for TMB values lower than 10. The input data included variables such as aluminum concentration (Al), immune checkpoint markers (CTLA-4, PD-1, PD-L1), cytokine levels (IFNγ), and immune cell populations. The data were standardized using z-scores to ensure that all variables contributed equally to the principal component calculation. The PCA was performed to reduce dimensionality while retaining key patterns, focusing on the first three principal components. A 3D biplot was generated to visualize the distribution of the samples across the first three principal components and illustrate the relationship between variables. Key variables contributing significantly to the differentiation between high and low TMB tissues were represented as vectors in the biplot. The placement and length of the vectors indicate the direction and magnitude of their influence.

## 5. Conclusions

This study demonstrates the presence of Al in all the investigated CRC samples. Nevertheless, the data reported here indicate that some hallmarks of cancer, such as TMB and the expression of immune response biomarkers, are strictly associated to Al concentration within the tumor tissue.

These findings open interesting prospects for the management of patients affected by CRC with high Al bioaccumulation, suggesting that high TMB, along with the upregulation of immune checkpoints, represents an ideal scenario for the application of immunotherapies. In this context, limiting environmental exposure to Al could potentially influence CRC progression by achieving better waste management practices, stricter regulations on industrial emissions, and increased monitoring of Al levels, mainly in food. Encouraging the use of Al-free personal care products and reducing the consumption of processed foods can all contribute to lowering Al exposure. In addition to regulatory and public health measures, further research is necessary to fully understand the mechanisms by which Al influences cancer progression and immune escape.

## Figures and Tables

**Figure 1 ijms-25-13388-f001:**
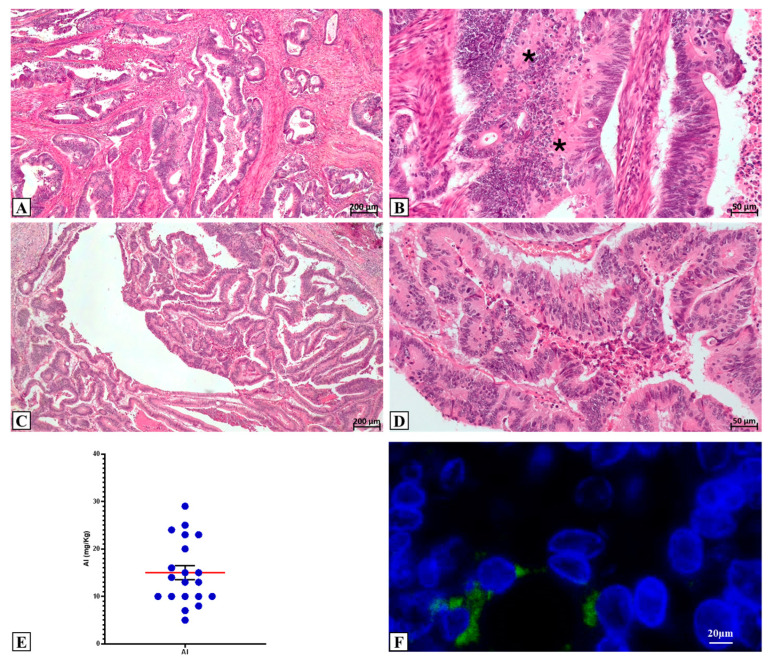
Histological analysis and aluminum detection. (**A**) Hematoxylin and eosin staining shows a colorectal adenocarcinoma with high inflammatory infiltrate. (**B**) High magnification of panel (**A**) displays numerous inflammatory cells (asterisk) next to cancerous ones. (**C**) Hematoxylin and eosin staining shows a colorectal adenocarcinoma. (**D**) High magnification of panel (**C**). (**E**) Aluminum concentration detected by ICP-MS analysis. (**F**) Colon cancer cells with aluminum (green; morin staining) in the cytoplasm. Red line represents the mean value.

**Figure 2 ijms-25-13388-f002:**
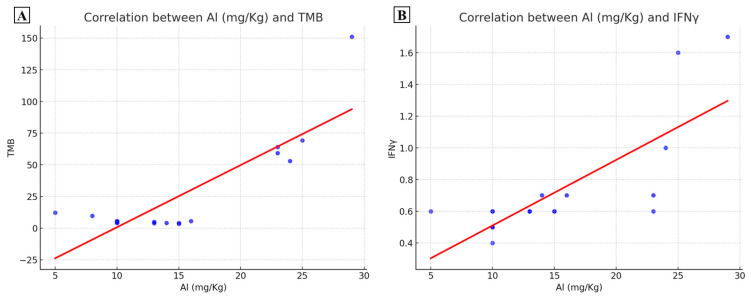
Effect of aluminum concentration on Tumoral mutational burden (TMB) and IFNγ. (**A**) Graph shows a positive association between aluminum concentration and TMB values. (**B**) Graph displays a positive association between aluminum concentration and IFNγ expression.

**Figure 3 ijms-25-13388-f003:**
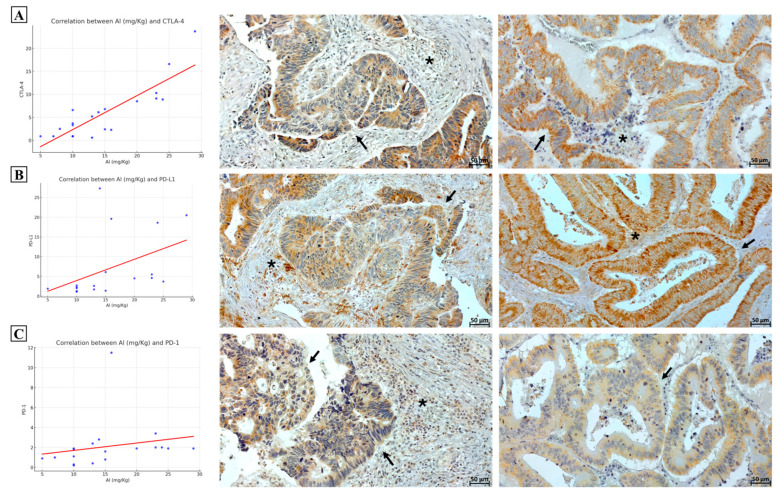
Effect of aluminum concentration on the expression of immune checkpoints. (**A**) Positive association between aluminum concentration and CTLA-4 expression (RNASeq). Immunohistochemistry shows CTLA-4 expression in both cancer cells (arrow) and inflammatory cells (asterisk). (**B**) Graph displays a positive trend between aluminum concentration and PD-L1 expression (RNASeq). Immunohistochemistry shows PD-L1 expression in both cancer cells (arrow) and inflammatory cells (asterisk). (**C**) Graph displays a positive trend between aluminum concentration and PD-1 expression (RNASeq). Immunohistochemistry shows PD-1 expression in both cancer cells (arrows) and inflammatory cells (asterisk).

**Figure 4 ijms-25-13388-f004:**
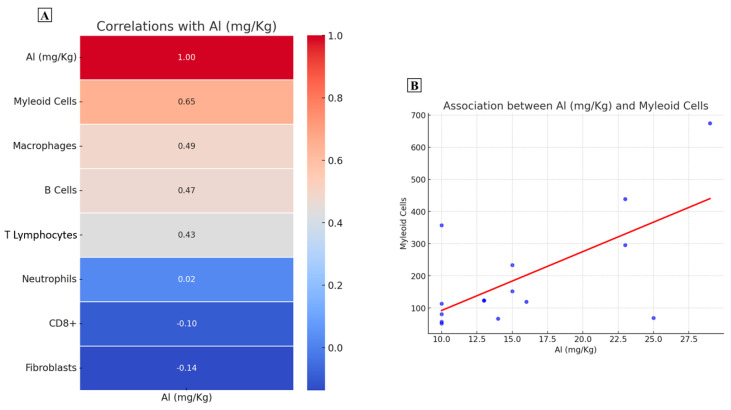
Association between aluminum concentration and immune cell types. (**A**) The heatmap reports the Pearson correlation values for the association between aluminum and the immune cell types. (**B**) Graph shows a positive association between aluminum concentration and myeloid cells.

**Figure 5 ijms-25-13388-f005:**
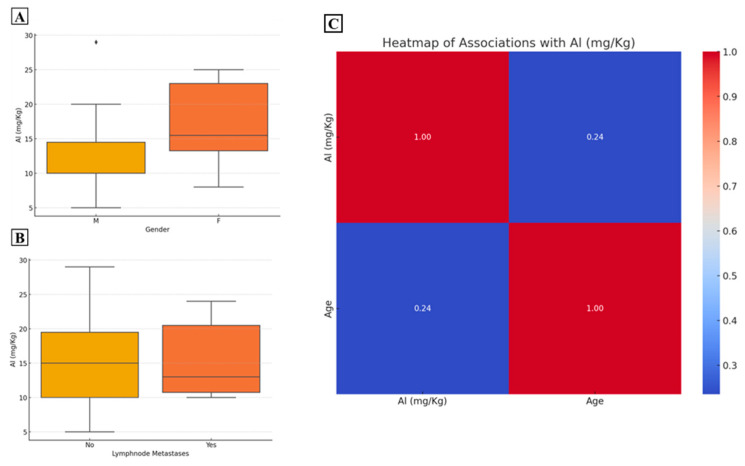
Association between aluminum concentration and categorical and continuous variables. (**A**) Graph shows the different aluminum concentration in male and female groups. (**B**) Graph displays the aluminum concentration in patients without and with lymph node metastasis at the time of surgery. (**C**) The heatmap reports the Pearson correlation values for the association between aluminum and age.

**Table 1 ijms-25-13388-t001:** Clinical and molecular characteristics of patients.

Al (mg/kg)	Age	Gender	Grade	Lymph Node Metastases	CTLA-4 TPM	PD-1 TPM	PD-L1 TPM	IFNγ TPM	TMB
29	82	M	High Grade	No	23.7	1.9	20.5	1.7	151.0
25	83	F	High Grade	No	16.6	1.9	3.7	1.6	69.3
24	74	F	High Grade	Yes	8.9	2.0	18.6	1.0	53.0
23	85	F	High Grade	No	9.1	3.4	5.5	0.7	59.3
23	61	F	Low Grade	Yes	10.3	2.0	4.6	0.6	64.1
20	84	M	High Grade	No	8.5	1.9	4.5	/	/
16	66	F	High Grade	No	2.3	11.5	19.6	0.7	5.5
15	76	M	High Grade	No	2.4	0.8	1.4	0.6	3.4
15	77	F	High Grade	No	6.8	1.6	6.1	0.6	4.0
14	87	F	High Grade	No	6.1	2.8	27.3	0.7	4.0
13	87	M	High Grade	Yes	0.6	0.4	1.7	0.6	4.9
13	85	F	High Grade	Yes	5.2	2.4	2.6	0.6	3.9
10	81	M	High Grade	No	3.7	1.9	2.3	0.6	4.2
10	73	M	High Grade	Yes	3.3	1.1	1.3	0.5	5.6
10	37	F	High Grade	Yes	6.6	1.8	2.7	0.6	5.6
10	48	M	High Grade	No	0.9	0.3	1.1	0.4	4.0
10	75	M	High Grade	No	0.9	0.2	2.0	0.5	4.7
8	85	F	High Grade	Yes	2.5	/	/	/	9.7
7	66	M	Low Grade	No	0.9	1.0	/	/	/
5	77	M	High Grade	No	0.9	0.9	1.9	0.6	12.2

TPM = transcript per million.

**Table 2 ijms-25-13388-t002:** Mutations in the main hallmark genes.

Al (mg/kg)	Somatic Mutations
29	ADGRA2, AGO1, AKT2, AKT3, AMER1, APOBEC3B, APOBEC3F, APOBEC3G, APOBEC4, AR, ARHGAP35, ARID1A, ARID2, ARID5B, ASXL1, ATRX, BCL9, BCORL1, BCR, BRD4, CARD11, CASP8, CBL, CD22, CREBBP, CTCF, CTNNB1, DAXX, DDR1, DDX4, DICER1, DIS3, EPCAM, EPHA5, EPHB2, ETV1, FAT1, FBXW7, FGF12, FGF4, FH, FLI1, GAB2, GATA3, GLI1, GNA13, GNAS, GRIN2A, GRM3, GTF2I, HERC2, HIF1A, HLA-B, HLA-C, HNF1A, IGF1, IKBKE, IL3, INPP4A, IRS1, JAK2, JAK3, JARID2, KDM5C, KLHL6, KMT2A, KMT2B, KMT2C, KMT2D, KSR2, LATS2, LMO2, LRP1B, LRP5, LZTR1, MAGI2, MAP3K4, MAX, MECOM, MEF2B, MGA, MGAM, MITF, MPL, MSI1, MSLN, MYOD1, NCOA3, NOTCH1
25	BCL2, MALT1, SETBP1
24	/
23	HLA-B, HLA-C, PTPN1
23	APOBEC3B, AR, ARHGAP35, ASXL1, AXIN1, CBL, CDC73, CENPA, CIC, CREBBP, CSNK1A1, CUX1, CYLD, ERG, EZH1, FANCF, FGF3, GATA3, GATA4, GRIN2A, INPP4B, KDM5A, KLF5, KMT2C, LRP5, MAD2L2, MED12, MITF, MYCN, NCOA3, NOTCH4, PBRM1, PGBD5, PIK3C3, POLD1, PPM1D, PPP2R1A, PREX2, PTPN11, PTPRS, RICTOR, RNF43, RPS6KA4, SS18, TAF1, TAL1, TCF7L2, TFE3, TGFBR2, TTK, YAP1, ZBTB20
20	/
16	/
15	APC, ATM, BIRC3, CCNE1, CHEK1, DDX4, EED, ETS1, FGF2, FLI1, FUBP1, IRF2, KAT6A, MRE11, NEGR1, PGR, PIK3R1, PLK2, SDHD, SMAD4, SPTA1, TCF7L2, YAP1
15	YY1
14	ADGRA2, ANKRD26, AR, ARID1A, ARID1B, ARID5B, ASXL1, ATXN7, BCORL1, BRD4, CHD2, CIC, CREB1, DICER1, DNMT3A, DOT1L, EPHB2, ERBB3, ETV4, FAT1, FBXW7, FGF8, FOXO1, GLI1, GNAS, HDAC4, HLA-A, HLA-B, HLA-C, IGF2, INSR, JAK3, KAT6A, KLF5, KMT2B, KMT2D, LRP1B, LRP6, MAF, MECOM, MGA, MST1R, NADK, NCOR1, NF1, NOTCH2, NOTCH3, NOTCH4, NRG1, NSD1, NTRK2, NTRK3, P2RY8, PAX5, PAX8, PHOX2B, PIK3C2G, PLCG1, PML, PPP2R2A, PTEN, PTPRS, PTPRT, QKI, RBM10, RECQL4, RNF43, ROCK1, RRAS, SETBP1, SMC1A, SOCS1, TERT, TTK, ZBTB20, ZFHX3
13	/
13	/
10	AKT1, ASXL1, CD74, DDX41, PAK5, PDGFRB, PLCG1, PTPRT, TOP1, U2AF1
10	BRCA2, CCND3, CDK8, CUL4A, CYSLTR2, DIS3, ERCC5, FGF14, FGF9, FLT1, FLT3, FOXO1, IRS2, KLF5, LATS2, PREX2, RB1, UBR5, VEGFA
10	BCL2L2, CDK6, CUX1, EPHB4, FAM46C, FGF17, FGF20, GATA4, KEL, NKX3-1, PARP2, PPP2R2A, SH2D1A, STAG2, TCF7L2, XIAP
10	BRCA2, CCNB3, CDK8, CUL4A, CYSLTR2, DIS3, ERCC5, FGF14, FGF9, FLT1, FLT3, FOXO1, IRS2, KLF5, LAMP1, LATS2, PLCG2, RANBP2, RB1, ZFHX3
10	APC, ARFRP1, CD70, DNMT1, ETV6, FBXW7, GNAS, HDAC7, INSR, KAT6A, LRP1B, MUC1, NCOA3, PTPN1, PTPRS, PTPRT, SMARCA4, SPOP, SRC, TBX3, TP53, VAV1, ZNF217
8	ABCB1, ADGRA2, APC, APOBEC1, ARHGEF28, ARID1A, ARID1B, ARID2, ASXL1, BCR, BRD4, CCND3, CCNE1, CDC73, CDH1, CDKN2A, CHD2, CIC, CNOT9, DEK, DHX15, DICER1, DNMT1, DOT1L, EP300, EPHA7, EPHB1, EPHB4, FBXW7, FGF13, FOXL2, FRS2, GAB1, GATA6, GNAS, GTF2I, HERC2, HLA-B, HLA-C, HNF1A, HSD3B1, INHA, JARID2, KMT2A, KMT2C, KMT2D, LRP1B, LRP5, LRP6, MALT1, MAP3K1, MEF2B, MGA, MLLT3, MSI1, MST1, NADK, NCOA3, NCOR1, NCSTN, NFKBIA, NRG1, NSD3, NUP98, NUTM1, PDCD1LG2, PPM1D, PPP2R1A, PREX2, PTCH1, PTPRS, PTPRT, RANBP2, RBM15, RNF43, ROCK1, RPS6KA4, RUNX1, SETBP1, SLX4, SMO, SOX9, SPEN, STK33, SUZ12, TBX3, TCF7L2, TERT, TGFBR2, TNFAIP3
7	/
5	ASXL1, ATRX, BCL2L1, BCR, CIC, CTNNB1, CXCR4, DNMT3B, GNAS, MAFB, MAGI2, MYC, NCOA3, PGR, PLCG1, PTPRT, RPS6KB2, SRC, TOP1, TP53, ZBTB7A

**Table 3 ijms-25-13388-t003:** Aluminum concentration and immune cell subtype reconstitution by deconvolution algorithm.

Al (mg/kg)	B Cells	Macrophages	T Lymphocytes	CD8+	Fibroblasts	Myleoid Cells	Neutrophils
29	80,955.8	3623.0	356.2	155.0	48,825.3	674.5	1952.5
25	4473.1	723.7	304.1	28.0	38,798.8	68.8	919.5
23	5648.6	2186.0	458.3	183.0	48,007.4	439.0	1033.0
23	18,969.9	1755.3	514.7	160.0	61,232.3	295.5	944.7
16	7234.7	1832.6	402.5	174.0	68,918.3	119.7	870.2
15	2281.9	642.4	246.3	86.0	15,741.5	233.2	1114.9
15	1430.7	1901.7	325.4	52.0	119,353.3	152.2	1146.1
14	1130.4	1849.1	244.1	66.0	120,199.5	67.0	1142.3
13	23,662.8	1174.7	422.4	129.0	60,896.8	124.2	1763.3
13	1354.2	2031.9	132.5	51.0	60,329.8	123.3	786.5
10	475.9	441.0	231.7	28.0	10,863.5	51.8	769.5
10	44,244.7	2349.4	414.3	180.0	177,744.3	357.2	1992.2
10	954.7	761.4	110.9	580.0	43,668.9	57.0	898.6
10	5423.1	1264.9	414.3	95.0	27,647.9	113.7	1464.9
10	646.1	1215.6	162.4	40.0	39,723.9	81.0	1432.3

## Data Availability

The original contributions presented in this study are included in the article. Further inquiries can be directed to the corresponding authors.

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
