# Peer review of "Aluminum Concentration Is Associated with Tumor Mutational Burden and the Expression of Immune Response Biomarkers in Colorectal Cancers"

_ijms, 2024, doi:10.3390/ijms252413388_

Round 1
Reviewer 1 Report
Comments and Suggestions for Authors
The manuscript by Bonfiglio et al. suggests the role of Al bioaccumulation in CRC progression while modulation inflammatory and immune response. Authors have provided a correlation analysis of Al with immune checkpoint markers and tumor mutational burden to support their hypothesis. However, major queries remain in the context of bioaccumulation of Al and colorectal cancer. Major comments are listed below:
1. The introduction section should be more focused with proper citations. Few citations do not match with given statement.
2. The authors should provide data for matched normal tissue compared to tumor tissue across all datasets.
3. Did the authors identify any specific somatic mutations associated with high levels of Al accumulation?
4. The authors should segregate patients based on MSI and MSS CRC tumor types, as this may provide crucial insights when comparing neoantigens.
5. The authors should perform a more detailed analysis of the correlation between major immune cell types and Al bioaccumulation.
6. Please include multiple representative H&E and IHC images. Additionally, provide low-magnification images.
7. In Figure 3, how do the authors explain the association between Al and immune checkpoint markers?
8. In Figure 2B, IFN-gamma could originate from multiple cell types. Can the authors specify whether its correlation with Al is linked to CD8, CD4, or other immune cells?
9. In Figure 3, the authors mention that increased IFN-gamma correlates with greater Al bioaccumulation. While IFN-gamma typically supports anti-tumor immunity and tumor regression, the evidence provided also suggests increased PD1 and PD-L1 expression correlating with Al accumulation, implying immune suppression. How do the authors reconcile these seemingly contradictory findings?
10. The authors should validate immune and cytokine markers using flow cytometry.
11. The manuscript title suggests that Al concentration is associated with cancer immune evasion. However, the authors have not provided sufficient evidence to support this conclusion.
12. The authors mention an association between Al and tumor mutational burden, which was previously published in Science of the Total Environment (doi.org/10.1016/j.scitotenv.2023.168335). Were the patient samples in this manuscript derived from the same cohort? Please clarify the novelty of this study compared to the earlier publication.
13. The authors should include in vivo validation to support the direct correlation between Al and immune checkpoint markers.
14. Reference 18 authors states, "Al can induce an immunosuppressive microenvironment by dysregulating immune checkpoint expression, thus supporting the survival, proliferation, and invasion of cancer cells." However, the cited reference does not mention Al. How did the authors draw this conclusion?
15. The manuscript’s conclusions appear hypothetical and lack strong justification or evidence.
16. The conclusion states, "The ability of Al to modulate inflammatory and immune responses highlights its role in cancer progression and its potential as a target for immunotherapy." However, the correlation between Al and immune checkpoint markers alone does not strongly support the claim that Al modulates immune responses in cancer progression.
Comments on the Quality of English Language
The overall quality of the manuscript should be improved with language editing services
Author Response
Manuscript ID ijms-3315462
"Aluminum Concentration Is Associated with Tumor Mutational Burden and The Expression of Immune Response Biomarkers in Colorectal Cancers"
Submitted to: International Journal of Molecular Science
ROUND#1
POINT-TO-POINT REBUTTAL TO REVIEWER COMMENTS
General Comments to Editor and Reviewers
We appreciated the thoughtful and constructive criticisms and suggestions of Reviewers. His/her comments on how to improve the manuscript, which has been revised accordingly. We also appreciate the Editors for calling for a new re-submission of an improved version of our manuscript.
REVIEWER#1
The manuscript by Bonfiglio et al. suggests the role of Al bioaccumulation in CRC progression while modulation inflammatory and immune response. Authors have provided a correlation analysis of Al with immune checkpoint markers and tumor mutational burden to support their hypothesis.
Reply: We sincerely thank the reviewer for their valuable suggestions, which have significantly improved the quality of our manuscript. We appreciate your thoughtful feedback on our study.
However, major queries remain in the context of bioaccumulation of Al and colorectal cancer. Major comments are listed below:
- The introduction section should be more focused with proper citations. Few citations do not match with given statement.
Reply: Thanks for this point out. We changed several references in the introduction section.
- The authors should provide data for matched normal tissue compared to tumor tissue across all datasets.
Reply: Thank you for your comment. We would like to clarify that our work aims to identify the molecular characteristics of colorectal tumors in association with aluminum bioaccumulation, rather than investigating the capacity of aluminum to trigger the carcinogenic process. Several studies have already demonstrated aluminum bioaccumulation in normal tissues and its differences from neoplastic tissues. However, very few, if any, have employed an approach similar to ours. We specified this in the new version of our manuscript.
In particular, the manuscript was modified as follow:
Introduction
Regarding the accumulation of Al in colon tissues, an epidemiological study revealed no significant differences in its concentrations by analyzing trace elements in both healthy and CRC biopsies [19]. However, the Al levels were slightly higher in CRCs compared to normal ones. In this context, we previously detected Al accumulation in colorectal can-cer (CRC) biopsies using histopathological techniques [20], establishing a link between Al bioaccumulation and key biological processes associated with CRC progression, such as epithelial-to-mesenchymal transition (EMT) [21,22] and resistance to cell death [23-26]. Additionally, a multiomics analysis conducted on two samples further suggests a correlation between Al presence and the occurrence of DNA mutation events. The presence of Al in CRC, which pathogenesis is rather complex [27-30], could be related to high frequence of Al contamination of food; therefore, intestinal absorption could be considered as a primary route of Al exposure, mainly considering its bioaccumulation in the gastrointestinal tract.
- Did the authors identify any specific somatic mutations associated with high levels of Al accumulation?
Reply: Although the Al concentration appears to be positively associated with the TMB, the mutational analysis did not reveal recurrent mutations in specific genes. We have clarified this point in the manuscript; see 2.2. Mutational State and Al Accumulation paragraph.
2.2. Mutational State and Al Accumulation
The possible mutational effect of Al bioaccumulation has been evaluated by corre-lating the Al levels with the TMB of colon cancer lesions. Figure 2A showed that results from linear regression analysis indicated a significative positive correlation with TMB (p < 0.0001; R-squared = 0.86). The mutational analysis also showed numerous somatic mutations in hallmark genes in the patient with the higher Al concentration (Table 2). Although the Al concentration appears to be positively associated with the TMB, the mutational analysis did not reveal recurrent mutations in specific genes.
- The authors should segregate patients based on MSI and MSS CRC tumor types, as this may provide crucial insights when comparing neoantigens.
Reply: we agree with the reviewer about the importance of MSI and MSS in CRC molecular classifications. However, the analyzed cohort (20 biopsies) is not sufficient to allow further subdivision of the populations for additional matching. We appreciate your suggestion and recognize its potential value for similar studies conducted on larger cohorts.
- The authors should perform a more detailed analysis of the correlation between major immune cell types and Al bioaccumulation.
Reply: Thanks for these important suggestions. In the new version of our manuscript, we added immune cell deconvolution analysis based on RNASeq data on a subpopulation of 15 samples. Specifically, immune cell type analysis is based on RNA-Seq data and was determined using MCPCounter (see Becht, E., et al. Genome Biol 17(218), 2016). This analysis allowed to associate Al concentration with immune cell infiltration. Table 3 and heatmap (Figure 4) showed a appositive association between Al concentration and myeloid cells. This association further supports the reported data. In fact, it is known that myeloid cells have immunosuppressive effects not only supporting immune escape directly by inducing immune check point expression, but also promoting tumor invasion via various non-immunological activities. We reported and discussed this data in the new version of our manuscript.
- Please include multiple representative H&E and IHC images. Additionally, provide low-magnification images.
Reply: we improve figure 1 and figure 3 by adding others H&E and IHC images.
- In Figure 3, how do the authors explain the association between Al and immune checkpoint markers?
Reply: We provide a possible explanation for this correlation in the discussion section, where we relate aluminum concentration to both IFNγ levels and immune checkpoints (see response to point #9).
- In Figure 2B, IFN-gamma could originate from multiple cell types. Can the authors specify whether its correlation with Al is linked to CD8, CD4, or other immune cells?
Reply: as reported in the response of point 5#, In the new version of our manuscript, we added immune cell deconvolution analysis based on RNASeq data.
- In Figure 3, the authors mention that increased IFN-gamma correlates with greater Al bioaccumulation. While IFN-gamma typically supports anti-tumor immunity and tumor regression, the evidence provided also suggests increased PD1 and PD-L1 expression correlating with Al accumulation, implying immune suppression. How do the authors reconcile these seemingly contradictory findings?
Reply: Thank you for this insightful comment. We acknowledge the apparent contradiction between the pro-inflammatory, anti-tumor role of IFNγ and the immune-suppressive environment suggested by the correlation between Al bioaccumulation and increased CTLA4 expression. This discrepancy can be explained by the highly pleiotropic nature of IFNγ, which exerts context-related effects depending on the cellular and molecular environment.
Although not mechanistic in nature, our findings suggest a consistent association between Al accumulation in CRC tissues and the development of an immunologically permissive tumor microenvironment. The observed correlation between IFNγ and Al, apparently paradoxical, may also align with this proposed scenario.
In fact, it is known that in the tumor microenvironment, IFNγ plays a dual role, regulating both antitumor and pro-tumorigenic immune responses. It functions as a cytotoxic cytokine, working alongside granzyme B and perforin to induce apoptosis in tumor cells [74]. At the same time, IFNγ promotes the expression of immune checkpoint and indoleamine-2,3-dioxygenase (IDO), thereby activating immune-suppressive pathways [75-79].
We added these considerations in the conclusions section.
- The authors should validate immune and cytokine markers using flow cytometry.
Reply: Thank you for your suggestion. This study was conducted on bioptic tissues, which are not suitable for flow cytometry analysis. However, we agree that such experiments would provide valuable insights, and we plan to include them in future studies to further validate immune and cytokine markers.
- The manuscript title suggests that Al concentration is associated with cancer immune evasion. However, the authors have not provided sufficient evidence to support this conclusion.
Reply: we changed the title with “Aluminum Concentration Is Associated with Tumor Mutational Burden and The Expression of Immune Response Biomarkers in Colorectal Cancers”
- The authors mention an association between Al and tumor mutational burden, which was previously published in Science of the Total Environment (doi.org/10.1016/j.scitotenv.2023.168335). Were the patient samples in this manuscript derived from the same cohort? Please clarify the novelty of this study compared to the earlier publication.
Reply: Thank you for your question. The previous study published in Science of the Total Environment was based on histopathological methodologies, with molecular investigations performed on only two cases, which revealed high tumor mutational burden (TMB) in patients with Al (no concentration data was reported). In contrast, this study represents a more comprehensive approach, utilizing ICP-MS for precise quantification of Al, RNA sequencing (RNASeq), and mutational analysis.
We specified this in the introduction section
In this context, we previously detected Al accumulation in colorectal cancer (CRC) bi-opsies using histopathological techniques [20], establishing a link between Al bioaccu-mulation and key biological processes associated with CRC progression, such as epithe-lial-to-mesenchymal transition (EMT) [21,22] and resistance to cell death [23-26]. Addi-tionally, a multiomics analysis conducted on two samples further suggests a correlation between Al presence and the occurrence of DNA mutation events. The presence of Al in CRC, which pathogenesis is rather complex [27-30], could be related to high frequence of Al contamination of food; therefore, intestinal absorption could be considered as a primary route of Al exposure, mainly considering its bioaccumulation in the gastroin-testinal tract. To further understand whether Al toxicity is dependent on its concentra-tion within samples, in the present work, we performed ICP-MS analysis to detect Al in 20 CRC biopsies subjected to both mutational and RNAseq analyses.
- The authors should include in vivo validation to support the direct correlation between Al and immune checkpoint markers.
Reply: We appreciate the reviewer’s suggestion, but in vivo validation goes beyond the scope of the present study. In fact, our work aims to associate the molecular characteristics of CRCs with Al levels. We have removed any mechanistic speculation from the manuscript, mainly in the “conclusion” section, to maintain focus on the proposed objectives.
- Reference 18 authors states, "Al can induce an immunosuppressive microenvironment by dysregulating immune checkpoint expression, thus supporting the survival, proliferation, and invasion of cancer cells." However, the cited reference does not mention Al. How did the authors draw this conclusion?
Reply: thanks for this point out. We changed the reference with “Zhu YZ, Liu DW, Liu ZY, Li YF. impact of aluminum exposure on the immune system: a mini review. Environ Toxicol Pharmacol. 2013 Jan;35(1):82-7. doi: 10.1016/j.etap.2012.11.009. Epub 2012 Nov 26.”
- The manuscript’s conclusions appear hypothetical and lack strong justification or evidence.
Reply: We have removed any mechanistic speculation from the manuscript, mainly in the “conclusion” section, to maintain focus on the proposed objectives.
However, in the new version of our manuscript we added immune cell deconvolution analysis based on RNASeq data on a subpopulation of 15 samples (see new Figure 4 and table 3) and a multivariate (see supplementary figure 1). This multivariate analysis permits a better understanding of the role of the different variables involved in the high TMB sample.
The following paragraphs were added to the text
Statistical analysis
Multivariate analyses were carried out with the Statistical software R (R version 4.2.1 (2022-06-23)). The sparse PLS – DA technique was applied at the aim of determining the combination of variables maximally discriminating between the tissues with high or low TMB levels. The sparse PLS - DA permits to projects the data in a subspace containing only the variables that explain the major fraction of variance. If the maximum number of variables is set, n, starting from an initial space dimension N, the principal components PC will be created as a linear combination of the n variables maximally contributing in explaining the variance. For the sparse PLS - DA a package of R in the library mixOmics was used. A dicotomic variable “TMB_index” was introduced charac-terized by two levels, 1 for TMB greater or equal to 10 and 0 for TMB values lower than 10.
The SPLS - DA was applied to the tissues at the aim of discriminating the sample with high or low TMB. The variables associated to the Al concentrations, the immune expression variables (i.e. CTLA_4, PD-1 and PD -L1), IFNγ, immune cells were included.
Results
Multivariate analysis further confirmed the association between Al concentration and high TMB. In fact, biplot graph (see supplementary figure 1) showed a prominent role of Al in discriminating high and low TMB samples, as evidenced by its strong loading on PC1 and its alignment with high TMB samples. Similar trend was observed for IFNγ and CTLA4 (supplementary figure 1).
- The conclusion states, "The ability of Al to modulate inflammatory and immune responses highlights its role in cancer progression and its potential as a target for immunotherapy." However, the correlation between Al and immune checkpoint markers alone does not strongly support the claim that Al modulates immune responses in cancer progression.
Reply: we removed this sentence in the new version of our manuscript.
Reviewer 2 Report
Comments and Suggestions for Authors
The study explores the correlation between aluminum bioaccumulation, tumor mutational burden (TMB), immune checkpoint molecule expression, and inflammatory markers in colorectal cancer (CRC). This research has significant public health implications given aluminum’s prevalence in consumer products and environment. This is undoubtedly an important topic. However, I would like to suggest several areas for improvement to enhance the manuscript.
Here are some suggestions for improving the manuscript:
1. Please verify the sample size, as it is referenced inconsistently as either 21 or 20 biopsies. The limited sample size reduces the reliability of the observed correlations.
2. Ensure consistent terminology throughout the manuscript, such as aluminum versus Al, IFN-γ versus IFN gamma, 4-µM versus four- µM.
3. In Figure 2 and 3, clearly label p value and R2 value in the correlation graphs and include units in the axis titles. In Figure 2B, adjust the layout so the y-axis title is fully visible.
4. The description of results for Figure 4 does not accurately reflect the data shown. Please review and revise to ensure accuracy.
5. Provide additional details in the Materials and Methods Section, such as microscopy information, wavelength, etc.
6. The study doesn’t include a control group (e.g., adjacent non-cancerous normal mucosa or healthy tissue samples) for comparison with the Al levels observed in CRC samples. Without this, it is challenging to conclude that the level of Al detected is uniquely elevated in cancerous tissue.
7. While this manuscript establishes correlations between Al concentration and TMB, as well as immune markers, it does not establish causation. It remains unclear how Al bioaccumulation induces these changes at molecular level. Experimental studies using in vitro or in vivo models would be necessary to confirm these causative links.
8. Consider examining environmental or dietary sources of aluminum exposure in relation to individual patient histories to strengthen the context of Al bioaccumulation.
9. Consider tracking how Al levels change over time or how they might correlate with disease progression or response to treatment.
Author Response
Manuscript ID ijms-3315462
"Aluminum Concentration Is Associated with Tumor Mutational Burden and The Expression of Immune Response Biomarkers in Colorectal Cancers"
Submitted to: International Journal of Molecular Science
ROUND#1
POINT-TO-POINT REBUTTAL TO REVIEWER COMMENTS
General Comments to Editor and Reviewers
We appreciated the thoughtful and constructive criticisms and suggestions of Reviewers. His/her comments on how to improve the manuscript, which has been revised accordingly. We also appreciate the Editors for calling for a new re-submission of an improved version of our manuscript.
REVIEWER#2
The study explores the correlation between aluminum bioaccumulation, tumor mutational burden (TMB), immune checkpoint molecule expression, and inflammatory markers in colorectal cancer (CRC). This research has significant public health implications given aluminum’s prevalence in consumer products and environment. This is undoubtedly an important topic.
Reply: We sincerely thank the reviewer for their valuable suggestions, which have significantly improved the quality of our manuscript. We appreciate your thoughtful feedback on our study.
However, I would like to suggest several areas for improvement to enhance the manuscript.
Here are some suggestions for improving the manuscript:
- Please verify the sample size, as it is referenced inconsistently as either 21 or 20 biopsies. The limited sample size reduces the reliability of the observed correlations.
Reply: Thanks for this point out. According to table 1, we analysed 20 samples from CRC patients. We reported this in the text.
- Ensure consistent terminology throughout the manuscript, such as aluminum versus Al, IFN-γ versus IFN gamma, 4-µM versus four- µM.
Reply: We have addressed this comment by ensuring consistent terminology throughout the manuscript.
- In Figure 2 and 3, clearly label p value and R2 value in the correlation graphs and include units in the axis titles. In Figure 2B, adjust the layout so the y-axis title is fully visible.
Reply: we changed all graphs.
- The description of results for Figure 4 does not accurately reflect the data shown. Please review and revise to ensure accuracy.
Reply: done. This is the new figure legend
Figure 5 (ex figure 4). Association between Aluminum concentration and categorical and continuous variables. (A) Graph shows the different aluminum concentration in male and female groups. (B) Graph displays the aluminum concentration in patients without and with lymph nodes metastasis at the time of surgery. (C) The heatmap reports the Pearson correlation values for the association between aluminum and all continuous variables evaluated in the study.
- Provide additional details in the Materials and Methods Section, such as microscopy information, wavelength, etc.
Reply: done
- The study doesn’t include a control group (e.g., adjacent non-cancerous normal mucosa or healthy tissue samples) for comparison with the Al levels observed in CRC samples. Without this, it is challenging to conclude that the level of Al detected is uniquely elevated in cancerous tissue.
Reply: Thank you for your comment. We would like to clarify that our work aims to identify the molecular characteristics of colorectal tumors in association with aluminum bioaccumulation, rather than investigating the capacity of aluminum to trigger the carcinogenic process. Several studies have already demonstrated aluminum bioaccumulation in normal tissues and its differences from neoplastic tissues. However, very few, if any, have employed an approach similar to ours. We specified this in the new version of our manuscript.
In particular, the manuscript was modified as follow:
Introduction
Regarding the accumulation of Al in colon tissues, an epidemiological study revealed no significant differences in its concentrations by analyzing trace elements in both healthy and CRC biopsies [19]. However, the Al levels were slightly higher in CRCs compared to normal ones. In this context, we previously detected Al accumulation in colorectal can-cer (CRC) biopsies using histopathological techniques [20], establishing a link between Al bioaccumulation and key biological processes associated with CRC progression, such as epithelial-to-mesenchymal transition (EMT) [21,22] and resistance to cell death [23-26]. Additionally, a multiomics analysis conducted on two samples further suggests a correlation between Al presence and the occurrence of DNA mutation events. The presence of Al in CRC, which pathogenesis is rather complex [27-30], could be related to high frequence of Al contamination of food; therefore, intestinal absorption could be considered as a primary route of Al exposure, mainly considering its bioaccumulation in the gastrointestinal tract.
- While this manuscript establishes correlations between Al concentration and TMB, as well as immune markers, it does not establish causation. It remains unclear how Al bioaccumulation induces these changes at molecular level. Experimental studies using in vitro or in vivo models would be necessary to confirm these causative links.
Reply: We appreciate the reviewer’s suggestion, but in vivo validation goes beyond the scope of the present study. In fact, our work aims to associate the molecular characteristics of CRCs with Al levels. We have removed any mechanistic speculation from the manuscript, mainly in the “conclusion” section, to maintain focus on the proposed objectives.
However, in the new version of the manuscript, we added immune cell deconvolution analysis based on RNASeq data. This analysis allowed to match Al concentration with immune cell infiltration. Table 3 and heatmap showed a positive association (r>0.6) only for the association between Al concentration and myeloid cells. This association further supports the reported data. In fact, it is known that myeloid cells have immunosuppressive effects not only supporting immune escape directly by inducing immune check point expression, but also promote tumor invasion via various non-immunological activities. We reported and discussed this data in the new version of our manuscript.
- Consider examining environmental or dietary sources of aluminum exposure in relation to individual patient histories to strengthen the context of Al bioaccumulation.
- Consider tracking how Al levels change over time or how they might correlate with disease progression or response to treatment.
Reply to point 8 and 9: Thank you for the suggestions. Currently, we do not have these data available; however, we are initiating studies to collect follow-up information and detailed occupational and environmental histories for the enrolled patients. These efforts aim to provide a more comprehensive understanding of aluminum exposure sources and its potential correlation with disease progression or treatment response in future research.
However, we extended the consideration about the possible dietary sources of aluminum.
Introduction
Due to its widespread use, Al exposure can occur via multiple routes including in-halation, ingestion through food and beverages, and dermal absorption. Al compounds are frequently used as food additives, such as stabilizers, colorants, and leavening agents, particularly in processed foods [11, 12]. Additionally, the use of Al cookware and utensils during food preparation can increase its content in meals, particularly when cooking acidic or salty foods. Beverages such as tea [13], which naturally contain Al, and Al-containing packaging materials further contribute to dietary exposure.
The daily intake of Al from food is estimated to range from 1 to 10 mg/day in the general population [14], though higher levels are observed in individuals consuming processed or packaged foods. This extensive exposure can lead to significant accumulation within the organism, potentially affecting health.
However, in the new version of our manuscript we added immune cell deconvolution analysis based on RNASeq data on a subpopulation of 15 samples (see new Figure 4 and table 3) and a multivariate (see supplementary figure 1). This multivariate analysis permits a better understanding of the role of the different variables involved in the high TMB sample.
The following paragraphs were added to the text
Statistical analysis
Multivariate analyses were carried out with the Statistical software R (R version 4.2.1 (2022-06-23)). The sparse PLS – DA technique was applied at the aim of determining the combination of variables maximally discriminating between the tissues with high or low TMB levels. The sparse PLS - DA permits to projects the data in a subspace containing only the variables that explain the major fraction of variance. If the maximum number of variables is set, n, starting from an initial space dimension N, the principal components PC will be created as a linear combination of the n variables maximally contributing in explaining the variance. For the sparse PLS - DA a package of R in the library mixOmics was used. A dicotomic variable “TMB_index” was introduced charac-terized by two levels, 1 for TMB greater or equal to 10 and 0 for TMB values lower than 10. The SPLS - DA was applied to the tissues at the aim of discriminating the sample with high or low TMB. The variables associated to the Al concentrations, the immune expression variables (i.e. CTLA_4, PD-1 and PD -L1), IFNγ, immune cells were included.
Results
Multivariate analysis further confirmed the association between Al concentration and high TMB. In fact, biplot graph (see supplementary figure 1) showed a prominent role of Al in discriminating high and low TMB samples, as evidenced by its strong loading on PC1 and its alignment with high TMB samples. Similar trend was observed for IFNγ and CTLA4 (supplementary figure 1).
Reviewer 3 Report
Comments and Suggestions for Authors
Dear authors,
I have now refereed the submission: In Situ Aluminum Concentration Is Associated with Tumor Mutational Burden and Cancer Immune Evasion in Colorectal Cancers. I have the following comments and suggestions.
1. The manuscript title could be revised to cover the aspect of ‘‘Inflammatory Responses’’ which appears silent in the current version. ‘‘In Situ’’ may not be necessary in the current title.
2. L19-20: Did the authors mean CRC patients? The statement seems to be incomplete.
3. The abstract needs to be rewritten so that it is in past tense. Please include the mean concentration and range of the Al analyzed.
4. For keywords, try as much as possible to only include words that are not part of the manuscript title. This way, the discoverability of the final published article can be increased.
5. L66-72: Please recheck these lines.
6. Under 4.6 (Statistical Analysis), where is the Pearson’s correlation analysis? Which software were used for the analysis?
7. Overall figure quality (especially graphs) are low, and there is misrepresentation of parameters. For example,
(a) In Figure 2b, the y-axis title is cut half-way. IFNγ is written as IFN_gamma which can cause confusion.
(b) In Figure 3, CTLA-4, PD-L1 and PD-1 were indicated as CTLA_4_T, PD_L1_T and PD_1_T, respectively.
(c) In Figure 1c, the information is not easily understandable. Sime other type of graphical representation could be better. Th x-axis has mg/kg while Al appears on the y-axis. What does the plot represent, and what does the middle line stands for?
8. Conclusions should not have citations.
9. Throughout the text, chose either Aluminum (in North American English) or Aluminium and use it but not both.
10. More information could be provided on the ICP-MS analysis for reproducibility. For instance,
(a) The detection limit for Al is not reported,
(b) Which gas was used and in which mode the analysis was run. Was there a specific isotope of Al considered?
(c) The analytical quality control and quality assurance procedures needs to be mentioned. ICP-MS analysis is frequently prone to contamination.
(d) What calibration standard was used and in which range?
Author Response
Manuscript ID ijms-3315462
"Aluminum Concentration Is Associated with Tumor Mutational Burden and The Expression of Immune Response Biomarkers in Colorectal Cancers"
Submitted to: International Journal of Molecular Science
ROUND#1
POINT-TO-POINT REBUTTAL TO REVIEWER COMMENTS
General Comments to Editor and Reviewers
We appreciated the thoughtful and constructive criticisms and suggestions of Reviewers. His/her comments on how to improve the manuscript, which has been revised accordingly. We also appreciate the Editors for calling for a new re-submission of an improved version of our manuscript.
REVIEWER#3
Dear authors,
I have now refereed the submission: In Situ Aluminum Concentration Is Associated with Tumor Mutational Burden and Cancer Immune Evasion in Colorectal Cancers. I have the following comments and suggestions.
Reply: We sincerely thank the reviewer for their valuable suggestions, which have significantly improved the quality of our manuscript. We appreciate your thoughtful feedback on our study.
- The manuscript title could be revised to cover the aspect of ‘‘Inflammatory Responses’’ which appears silent in the current version. ‘‘In Situ’’ may not be necessary in the current title.
Reply: thanks for this point out. We modified the title of our manuscript according to the reviewer suggestions.
Aluminum Concentration Is Associated with Tumor Mutational Burden and The Expression of Immune Response Biomarkers in Colorectal Cancers
- L19-20: Did the authors mean CRC patients? The statement seems to be incomplete.
Reply: we corrected this.
- The abstract needs to be rewritten so that it is in past tense. Please include the mean concentration and range of the Al analyzed.
Reply: We rewritten the abstract according to the reviewer suggestion and the new findings.
- For keywords, try as much as possible to only include words that are not part of the manuscript title. This way, the discoverability of the final published article can be increased.
Reply: thanks for this point out. These are the new kye words: aluminum; colorectal cancer; TMB; immune escape; immune checkpoint; IFNγ; Myeloid cells; environmental pollution.
- L66-72: Please recheck these lines.
Reply: done
- Under 4.6 (Statistical Analysis), where is the Pearson’s correlation analysis? Which software were used for the analysis?
Reply: we rewrite statistical analysis section according to both reviewer suggestion and new data.
Specifically, the section has been modified as follow:
4.6. Statistical Analysis
Pearson correlation analysis was performed to evaluate the linear relationship between Al and others continuous variables. The correlation coefficient (r) and corresponding p-value were calculated to determine the strength and statistical significance of associations. Analysis was conducted using Statistical software R (R version 4.2.1 (2022-06-23).
The differences in Al concentration between categorical variables were analyzed using a t-test. RNASeq data were reported as transcript per million (TPM). Al concentration was reported as mean ± SEM.
Multivariate analyses were carried out with the Statistical software R (R version 4.2.1 (2022-06-23)). The sparse PLS – DA technique was applied at the aim of determining the combination of variables maximally discriminating between the tissues with high or low TMB levels. The sparse PLS - DA permits to projects the data in a subspace containing only the variables that explain the major fraction of variance. If the maximum number of variables is set, n, starting from an initial space dimension N, the principal components PC will be created as a linear combination of the n variables maximally contributing in explaining the variance. For the sparse PLS - DA a package of R in the library mixOmics was used. A dicotomic variable “TMB_index” was introduced characterized by two levels, 1 for TMB greater or equal to 10 and 0 for TMB values lower than 10.
However, in the new version of our manuscript we added immune cell deconvolution analysis based on RNASeq data on a subpopulation of 15 samples (see new Figure 4 and table 3) and a multivariate (see supplementary figure 1). This multivariate analysis permits a better understanding of the role of the different variables involved in the high TMB sample.
The following paragraphs were added to the text
Statistical analysis
Multivariate analyses were carried out with the Statistical software R (R version 4.2.1 (2022-06-23)). The sparse PLS – DA technique was applied at the aim of determining the combination of variables maximally discriminating between the tissues with high or low TMB levels. The sparse PLS - DA permits to projects the data in a subspace containing only the variables that explain the major fraction of variance. If the maximum number of variables is set, n, starting from an initial space dimension N, the principal components PC will be created as a linear combination of the n variables maximally contributing in explaining the variance. For the sparse PLS - DA a package of R in the library mixOmics was used. A dicotomic variable “TMB_index” was introduced charac-terized by two levels, 1 for TMB greater or equal to 10 and 0 for TMB values lower than 10.
The SPLS - DA was applied to the tissues at the aim of discriminating the sample with high or low TMB. The variables associated to the Al concentrations, the immune expression variables (i.e. CTLA_4, PD-1 and PD -L1), IFNγ, immune cells were included.
Results
Multivariate analysis further confirmed the association between Al concentration and high TMB. In fact, biplot graph (see supplementary figure 1) showed a prominent role of Al in discriminating high and low TMB samples, as evidenced by its strong loading on PC1 and its alignment with high TMB samples. Similar trend was observed for IFNγ and CTLA4 (supplementary figure 1).
- Overall figure quality (especially graphs) are low, and there is misrepresentation of parameters. For example,
(a) In Figure 2b, the y-axis title is cut half-way. IFNγ is written as IFN_gamma which can cause confusion.
(b) In Figure 3, CTLA-4, PD-L1 and PD-1 were indicated as CTLA_4_T, PD_L1_T and PD_1_T, respectively.
(c) In Figure 1c, the information is not easily understandable. Sime other type of graphical representation could be better. Th x-axis has mg/kg while Al appears on the y-axis. What does the plot represent, and what does the middle line stands for?
Reply: We changed all graphs in the new version of our manuscript.
- Conclusions should not have citations.
Reply: we deleted citations from the discussion
- Throughout the text, chose either Aluminum (in North American English) or Aluminium and use it but not both.
Reply: done
- More information could be provided on the ICP-MS analysis for reproducibility. For instance,
(a) The detection limit for Al is not reported,
(b) Which gas was used and in which mode the analysis was run. Was there a specific isotope of Al considered?
(c) The analytical quality control and quality assurance procedures needs to be mentioned. ICP-MS analysis is frequently prone to contamination.
(d) What calibration standard was used and in which range?
Reply: we implemented the material and method section.
Round 2
Reviewer 2 Report
Comments and Suggestions for Authors
The authors have appropriately revised this manuscript. I recommend to accept it.
Author Response
Manuscript ID ijms-3315462
"Aluminum Concentration Is Associated with Tumor Mutational Burden and The Expression of Immune Response Biomarkers in Colorectal Cancers"
Submitted to: International Journal of Molecular Science
ROUND#2
POINT-TO-POINT REBUTTAL TO REVIEWER COMMENTS
General Comments to Editor and Reviewers
We appreciated the thoughtful and constructive criticisms and suggestions of Reviewers. His/her comments on how to improve the manuscript, which has been revised accordingly. We also appreciate the Editors for calling for a new re-submission of an improved version of our manuscript.
REVIEWER#2
The authors have appropriately revised this manuscript. I recommend to accept it.
Reply: We sincerely thank the reviewer for their valuable work, which have significantly improved the quality of our manuscript. We appreciate your thoughtful feedback on our study.
Reviewer 3 Report
Comments and Suggestions for Authors
The authors have answered most of my concerns. However, I see that:
(a) The ICP-MS methodology is still very scanty. Could you please check my previous comments and try to address them?
(b) The biplot provided in the supplementary materials has only upto about 55% of the variance explained. This may imply that the 3rd component may explain more variance.
Author Response
REVIEWER#3
The authors have answered most of my concerns.
Reply: We sincerely thank the reviewer for their valuable work, which have significantly improved the quality of our manuscript. We appreciate your thoughtful feedback on our study.
However, I see that:
- The ICP-MS methodology is still very scanty. Could you please check my previous comments and try to address them?
Reply: Thanks for this point out. In the new version of our manuscript, we reported all information required by the reviewer.
(b) The biplot provided in the supplementary materials has only upto about 55% of the variance explained. This may imply that the 3rd component may explain more variance.
Reply: We agree with the reviewer. In the new version of our manuscript, we extended our analysis with the 3rd component. The 3D biplot has been added to the manuscript.
REVIEWER#3
The authors have answered most of my concerns.
Reply: We sincerely thank the reviewer for their valuable work, which have significantly improved the quality of our manuscript. We appreciate your thoughtful feedback on our study.
However, I see that:
- The ICP-MS methodology is still very scanty. Could you please check my previous comments and try to address them?
Reply: Thanks for this point out. In the new version of our manuscript, we reported all information required by the reviewer.
(b) The biplot provided in the supplementary materials has only upto about 55% of the variance explained. This may imply that the 3rd component may explain more variance.
Reply: We agree with the reviewer. In the new version of our manuscript, we extended our analysis with the 3rd component. The 3D biplot has been added to the manuscript.